# Addition of alkynes and osmium carbynes towards functionalized $d_\pi$–$p_\pi$ conjugated systems

Shiyan Chen[1,3], Longzhu Liu[2,3], Xiang Gao[1], Yuhui Hua [1], Lixia Peng[1], Ying Zhang[1], Liulin Yang[1], Yuanzhi Tan [1], Feng He [2✉] & Haiping Xia [1,2✉]

The metal-carbon triple bonds and carbon-carbon triple bonds are both highly unsaturated bonds. As a result, their reactions tend to afford cycloaddition intermediates or products. Herein, we report a reaction of M≡C and C≡C bonds that affords acyclic addition products. These newly discovered reactions are highly efficient, regio- and stereospecific, with good functional group tolerance, and are robust under air at room temperature. The isotope labeling NMR experiments and theoretical calculations reveal the reaction mechanism. Employing these reactions, functionalized $d_\pi$-$p_\pi$ conjugated systems can be easily constructed and modified. The resulting $d_\pi$-$p_\pi$ conjugated systems were found to be good electron transport layer materials in organic solar cells, with power conversion efficiency up to 16.28% based on the PM6: Y6 non-fullerene system. This work provides a facile, efficient methodology for the preparation of $d_\pi$-$p_\pi$ conjugated systems for use in functional materials.

[1] State Key Laboratory of Physical Chemistry of Solid Surfaces and Collaborative Innovation Center of Chemistry for Energy Materials (iChEM), College of Chemistry and Chemical Engineering, Xiamen University, 361005 Xiamen, China. [2] Shenzhen Grubbs Institute and Department of Chemistry, Southern University of Science and Technology, 518055 Shenzhen, China. [3]These authors contributed equally: Shiyan Chen, Longzhu Liu. ✉email: hef@sustech.edu.cn; xiahp@sustech.edu.cn

Compounds with transition metal–carbon triple bonds, namely metal carbyne complexes, have attracted much interest and attention since the seminal work by Fischer et al.[1–3]. The M≡C bonds have been shown rich reactivity with electrophiles and nucleophiles[4–8]. Therefore, metal carbyne complexes are valuable in synthetic chemistry as precursors for interesting organometallic compounds[9–11], and can be used as homogeneous catalysts in the synthesis of organic compounds and polymers[12–18]. Alkynes containing C≡C bonds are also a class of compounds with triple bonds. They are used as versatile synthetic intermediates and are ubiquitous among complex organic molecules and polymers with important applications[19–25].

The reactions of M≡C and C≡C bonds tend to afford cycloaddition intermediates or products on account of their high degree of unsaturation. For example, the alkyne metathesis undergoes metallacyclobutadiene intermediates process[12–14], and [2+2], [2+2+1], and [2+2+2] cycloaddition products have been widely published[26–37]. Acyclic products, however, have never been reported. Therefore, construction of acyclic compounds from compounds with these two highly unsaturated triple bonds remains great challenge.

Herein, we report a reaction between metal carbynes and terminal alkynes in the presence of acid, which gives rise to acyclic addition products. This kind of reaction is highly efficient because of the extreme angle strain associated with the small carbyne bond angle in the cyclic metal carbynes. Meanwhile, these reactions show regio- and stereoselectivity, delivering exclusively *trans* products and *anti*-Markovnikov addition reactions. In contrast to common organometallic reactions that are sensitive to air and temperature, metal carbynes possess good functional group tolerance for alkynes and are robust in air and under ambient conditions.

The M≡C bond in this work is cooperative in metalla-aromatic systems in which the transition metal *d* orbits and the carbon *p* orbitals overlap to form $d_\pi$–$p_\pi$ conjugated systems[38]. By taking advantage of this kind of reaction, the $p_\pi$–$p_\pi$ conjugation of the organic moiety and the $d_\pi$–$p_\pi$ conjugation of organometallics can be linked in a single conjugated system, which endows these unique acyclic addition products with potential applications especially in the field of optoelectronics. Because of the electron transport feature enhanced by osmium, these $d_\pi$–$p_\pi$ conjugated complexes have been proved to be feasible as electron transport layers (ETLs) to enhance device performance. For example, PM6: Y6-based non-fullerene solar cells with complex **30** as an ETL have acquired a power conversion efficiency (PCE) as high as 16.28%. This value is not only much higher than that in devices lacking ETL, but is also dramatically elevated compared with commonly used ETLs, such as the PDINO discussed below. This reveals that the $d_\pi$–$p_\pi$ conjugated complexes have great potential as interfacial layer materials, especially when combined with good electron transport capability and solubility in alcohol to facile orthogonal solvent processing, which could open a window into the capabilities of these newly developed $d_\pi$–$p_\pi$ conjugated complexes.

## Results

### Reactions of alkynes with osmapentalynes.
The osmapentalyne reactants such as **1a** (Fig. 1a) were synthesized and characterized (See Supplementary Figs. 40–43)[39]. A solution of the complex (**1a**) was treated with HCl·Et₂O solution in dichloromethane (DCM), producing a carbyne triple bond shifted product (**2a**) (Fig. 1a). The clear nuclear magnetic resonance (NMR) signal changes accompany this process (Supplementary Figs. 60–63). And the structures of **1a** and **2a** were confirmed by X-ray

diffraction. As shown in Fig. 1b, the Os1≡C1 triple bond length is 1.840 Å and the Os1–C7 single bond length is 2.061 Å in **1a**. In **2a** in contrast, the Os1≡C7 triple bond length is 1.855 Å and the Os1–C7 single bond length is 2.029 Å. These data indicate a tautomeric shift of the Os≡C triple bond from the initial ring to a different five-membered ring after the addition of acid. The similar tautomerization of the π-system from **1a** to **2a** has been discussed in detail in our previous work[40]. The Os1–C7–C6 metal-carbyne angle in **2a** is 128.6°, smaller than the corresponding angle in **1a** (130.9°). Both of them deviate by nearly 50° from the ideal *sp*-hybridized carbon angle (180°). The high ring strain associated with this small carbyne angle promoted us to investigate the reactivity of **1a** and **2a** with alkynes. In our previous work, only cycloaddition reactions based on similar osmapentalyne complexes were reported[27,30,31].

In this work, we discovered that the osmapentalyne (**2a**) reacts readily with phenylacetylene at room temperature (RT) in the presence of excess HCl·Et₂O, forming an acyclic addition complex (**3**) (Fig. 1). In view of the shifted carbyne triple bond results described above, we used complex **1a** as a reactant in an attempt to simplify the operation. As expected, the same product (**3**) was obtained within 5 min in 99% isolated yield, showing that complex **1a** can be a good precursor in this type of reaction. It is worth noting that the Os≡C triple bond of complex **2a** can be slowly oxidized in air and the reaction is improved in an inert atmosphere. In contrast, the Os≡C triple bond of precursor **1a** is more stable because of protection from the bulky triphenylphosphonium substituents. With precursor **1a** as a reactant, the reactions can be carried out smoothly on gram scale in air and with commercial solvents such as dichloromethane, acetonitrile or 1,2-dichloroethane. This is different from general organometallic reactions that require oxygen-free and anhydrous conditions. Therefore, we used complex **1a** as the reactant in the following investigations.

The structure of complex **3** was confirmed by X-ray single-crystal diffraction. As shown in Fig. 1b, it is an *anti*-Markovnikov addition product. Two building blocks, metallacycle and styrene, are connected by a C7–C12 bond of length 1.449 Å. The C12–C13 bond length of 1.357 Å suggests that it is a typical C=C double bond. The angles of C7–C12–C13 and C12–C13–C14 are 123.4° and 127.8°, respectively. The carbon–carbon bond lengths within the metallacycles (1.390–1.402 Å) are intermediate between typical single- and double-bond lengths. The Os1–C1 bond length (1.844 Å) in **3** is almost the same as the metal-carbyne bond length (1.840 Å) in complex **1a**, and the Os1–C1–C2 metal-carbyne angle is 130.0°.

Complex **3** was also characterized by NMR spectroscopy, elemental analysis (EA) and high-resolution mass spectrometry (HRMS) (Supplementary Figs. 68–71). Two singlets at 4.81 and -0.18 ppm, observed in the ³¹P NMR spectrum in CD₂Cl₂ can be assigned to C*P*Ph₃ and Os*P*Ph₃, respectively. In ¹H NMR spectrum, the two singlets at 7.44 and 5.87 ppm are assigned to C12*H* and C13*H*, respectively. The coupling constant of the newly formed *H*C12=C13*H* group is 16.90 Hz, confirming the *E*-isomer configuration[41]. No resonance peaks associated with branched and *Z*-isomers were observed, indicating the regio- and stereo-specifiity of this reaction. The ¹³C NMR spectrum in CD₂Cl₂ showed a signal from the carbyne carbon C1 at 316.2 ppm, and a signal from C7 at 218.8 ppm. Additionally, HRMS showed a peak at m/z 1347.3621, consistent with the theoretical structure of [$C_{79}H_{63}ClOOsP_3$]⁺ (1347.3387).

### Mechanism study.
To gain further insight into these reactions, we designed isotope labeling NMR experiments (Fig. 2a) in which DCl·Et₂O was used instead of HCl·Et₂O. We found that the

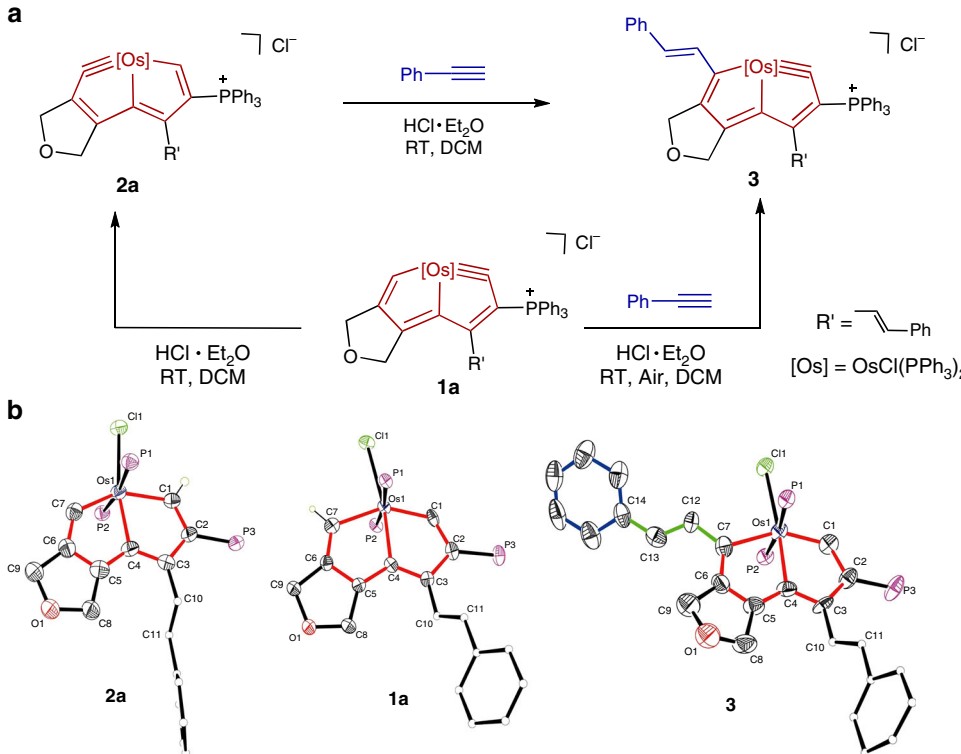

**Fig. 1 Reactions of alkynes with osmapentalynes. a** Reaction of alkynes and osmapentalynes and the reaction of carbyne triple bond shifted reaction. **b** X-ray molecular structure for the cations of complex **1a**, complex **2a**, and complex **3**. Ellipsoids are at the 50% probability level; phenyl groups in PPh₃ are omitted for clarity.

hydrogen on the *trans*-olefins formed (C13H) was from the added acid. Deuterated phenylacetylene was also tested in the reaction, and as expected, the hydrogen atom of terminal alkyne of phenylacetylene was not involved in the reaction. The identifications of **3**-$d_1$ and **3**-$d_2$ were supported by NMR spectroscopy and HRMS (Supplementary Figs. 204–211).

The carbene mechanism was proposed for the formation of a $\eta^3$-vinylcarbene ligand via the reaction between the alkyne and the protonated carbyne ligand[42]. We designed a verification experiment to determine whether this reaction follows the same mechanism. In our previous work, we demonstrated the conversion of the Os≡C bond in osmapentalyne to an Os=C bond upon addition of the strong acid, HBF₄·Et₂O[40]. Herein, we prepared the osmapentalene (**37**) containing an Os=C bond using the same method (Fig. 2b). The structure of the complex (**37**) was further confirmed by X-ray diffraction (Supplementary Fig. 7). However, treatment of **37** with phenylacetylene failed to produce complex **31′**. Indeed, **37** cannot be reacted with phenylacetylene. Therefore, a carbene intermediate in this reaction mechanism is excluded.

The participation of terminal alkynes in catalytic alkyne metathesis has been reported recently[43,44]. Then, we questioned if this kind of reaction involves a [2+2] cycloadditon intermediate which is the most common type of reaction between an Os≡C bond and a C≡C bond. We prepared the [2+2] cycloadditon complex (**38**) by reacting complex **2b** with phenylacetylene without any added acid (Fig. 2b). However, complex **38** is not converted to complex **31** notwithstanding addition of HCl·Et₂O or any other acid, indicating that the reactions fail to follow the [2+2] cycloaddition route.

Following these results, DFT calculations were carried out to shed light on the mechanism of these reactions. The computed Gibbs free energy profile of the key reaction steps is shown in Fig. 3. First, under the synergistic effect of acid, the Os≡C bond of

osmapentalyne (**2a**) reacts with the C≡C bond of phenylacetylene to form the Z-isomer intermediate (**Int1**), in which the metal center has only 16 electrons. This process has an energy barrier of 18.8 kcal mol⁻¹ and is exergonic by 47.6 kcal mol⁻¹. Secondly, the styryl group coordinates with the osmium center in an $\eta^2$ manner to form **Int2**, and subsequently converted into **Int3** with an energy barrier of 10.8 kcal mol⁻¹. Notably, the computed $\eta^2$-complexes (**Int2**, **Int3**) are resemblance to $\eta^2$-complexes of ruthenium, which showed as key intermediates of various unconventional addition reactions to alkynes[45]. Subsequently, the chloride anion approaches the hydrogen at C1 to form an E-isomer (**Int4**), further reducing the energy. Finally, **Int4** can easily lose the proton on C1 to form the osmapentalyne (**3**). This last step, from **Int4** to **3** has an energy barrier of 4.5 kcal mol⁻¹ and is exergonic by 4.6 kcal mol⁻¹. In the process, the metal center returns to a stabilized 18 electron structure. We have reported previously the possibility of conversion of a 16-electron osmapentaene to an 18-electron osmapentalyne[40]. The reaction is exremely fast and efficient and the 16-electron osmapentaene intermediate was undetectable. Protons from the acid play a particularly important role and the common reaction of Os≡C bond with the C≡C bond shows very unique results.

**Substrate scope.** To investigate the functional group tolerance of this kind of reaction, a broad species of alkynes and osmapentalyne complexes were investigated. With regard to the alkyne scope (Fig. 4), a series of substrates with either electron-withdrawing or electron-donating groups at different positions of their phenyl rings, perform well in this reaction, affording **3**-**21** with 90–99% isolated yields. All these reactions are essentially quantitative according to NMR data, and the slightly lower yields were mainly the result of purification steps necessitated by the highly polar triphenylphosphine substituents in the products.

**a** Deuterium labeling experiments

**b** Verification experiment

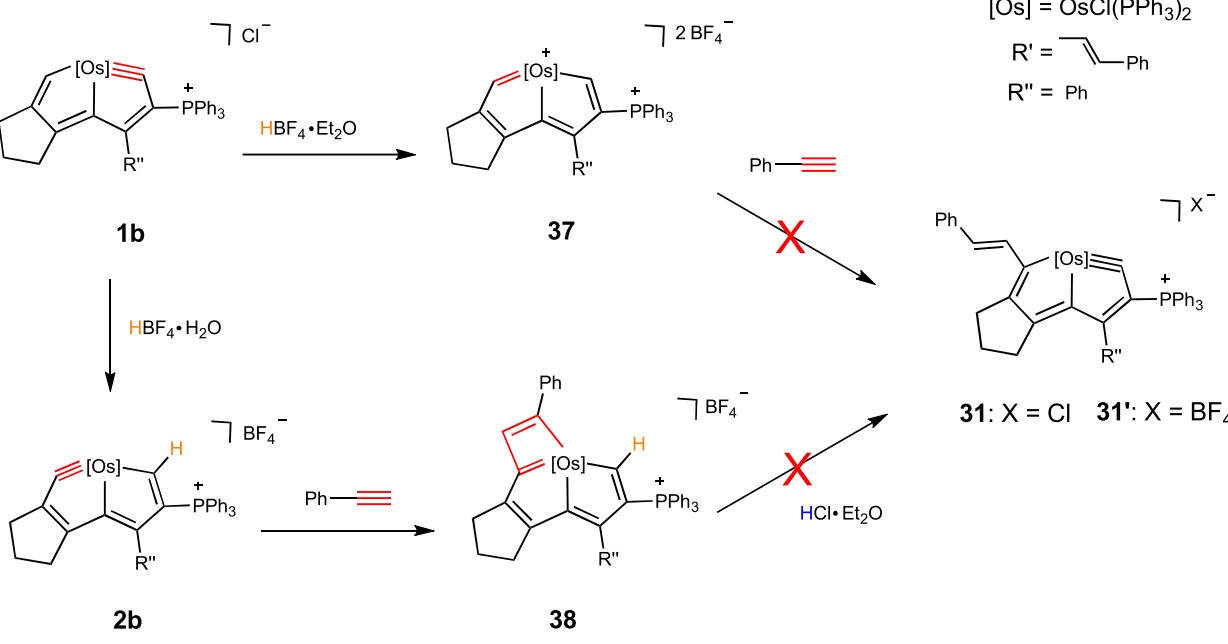

**Fig. 2 Mechanistic studies.** (**a**) isotope labeling NMR experiments and (**b**) verification experiment.

Common substituents, such as halogen (**5-11**), terminal olefin (**12**), methoxylcarbonyl (**13**), carboxyl (**14**), hydroxyphenyl (**15**), nitro (**16**), methoxyl (**17**), free amino (**18**), cyano (**19**), formyl (**20**), and pinacolborato (**21**) are all compatible. In addition to substituted phenyl groups, commercially available alkynes containing heterocycles such as thiophene (**22**), benzo[*d*]thiazole (**23**), pyridine (**24**), pyrimidine (**25**), quinoline (**26**) and imidazo [1,2-*b*]pyridazine (**27**) also provide the desired products in good isolated yields (85–99%). For non-aromatic substituted alkynes, we found only methyl propiolate (**28**) and propiolic acid (**29**) can undergo the reaction efficiently (See Supplementary Figs. 168–175).

Concerning the scope of the osmapentalyne (Fig. 4), a series of reactant osmapentalynes (**1b-1g**) were synthesized by the treatment of carbon ligands (**L2-L7**) with multiyne chains with OsCl$_2$(PPh$_3$)$_3$ and PPh$_3$, the different substituents at different positions appear not to affect the reaction (**3**, **31-36**). For instance, the R′ substituent can be styryl (**3**), phenyl (**31-33**), thienyl (**34**) or hydrogen (**35-46**), and the Y group can be CH$_2$ (**31**, **34**), O (**32**, **35**) or C(COOMe)$_2$ (**33**, **36**). The structures of all the above complexes (**3-36**) were confirmed by NMR spectroscopy, EA and HRMS (Supplementary Figs. 68–203). Some of the structures, such as **15** and **29** were further confirmed by X-ray diffraction (Supplementary Figs. 4 and 5). Based on these results,

we were able to conclude that these reactions adapt well to substituents and tolerate functional groups well.

**Construction of large $d_\pi$–$p_\pi$ conjugated systems**. Significantly, the Os≡C bond described here cooperates in $d_\pi$–$p_\pi$ conjugation metalla-aromatic systems in which the transition metal *d* orbits and the carbon *p* orbits have good overlap[38]. Thanks to the efficiency of these reactions, the $p_\pi$–$p_\pi$ conjugated organic systems and $d_\pi$–$p_\pi$ conjugated metallacycle systems can be easily connected through C–C bond coupling, and as a result, the $d_\pi$–$p_\pi$ conjugated systems could be further extended. Furthermore, the good group tolerance of this kind of reaction facilitates the molecular design of target products.

To prepare large π-systems containing $d_\pi$–$p_\pi$ conjugation, we treated the osmapentalyne complex (**1a**) with one equivalent of 1-ethynylpyrene in DCM, adding HCl·Et$_2$O for 15 min at RT in air (Fig. 5a). As expected, the complex (**30**) was isolated in 95% yield. The structure of **30** was characterized by NMR spectroscopy, EA and HRMS (Supplementary Figs. 176–179). The single-crystal X-ray diffraction of **30′**, a derivative of **30** with BF$_4^-$ as the counter anion, clearly revealed the acyclic geometry structure that is analogous to complex **30** (Fig. 5a).

The UV/Vis absorption spectra can reflect the extent of π-conjugation. As shown in Fig. 5b, no absorption peak in the

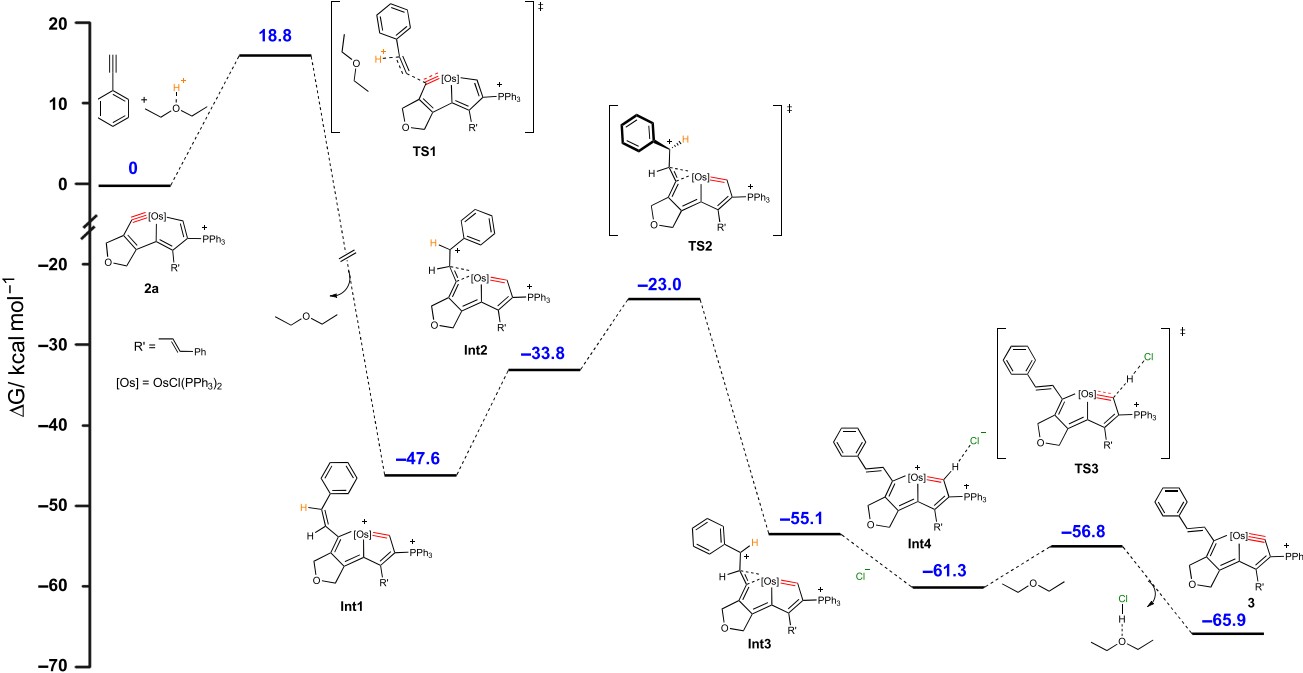

**Fig. 3 Gibbs free energy profile for the DFT-calculated possible mechanism.** The computed free energies are in kcal mol$^{-1}$.

region over 600 nm is detected from either the reactant aromatic alkynes or the osmapentalyne (**1a**). In contrast, the products **3** and **30** show strong absorption in the low-energy absorption region. In comparison with the absorption spectra of reactant **1a** and product **3**, the characteristic energy absorption band of **3** is redshifted from 465 nm to 572 nm with the extended π-conjugated framework. The molar absorption coefficient of product **3** at 572 nm is as high as $4.3 \times 10^4 \, M^{-1} \, cm^{-1}$ while that of the reactant **1a** is only $1.0 \times 10^4 \, M^{-1} \, cm^{-1}$ in the corresponding low-energy absorption region. With further extended *d–p* π-conjugation, the characteristic energy absorption band of **30** is further redshifted compared to complex **3**. It is noteworthy that complex **30** demonstrate broad and strong absorption in the entire visible region. Therefore, this kind of reaction provides a powerful method with which to construct large π-systems containing $d_\pi$–$p_\pi$ conjugation.

To gain insight into these intriguing absorption spectra, we carried out time-dependent DFT (TD-DFT) calculations on the cationic part of **3** (Supplementary Table 7). The $\lambda_{max} = 557$ nm of **3** is 96.7% ascribed to HOMO → LUMO transitions. This is a clear $d_\pi$–$p_\pi$ π-conjugation system demonstrated by the simulated distribution of HOMO and LUMO orbitals (Fig. 5c). There are strong π-delocalization between the *d* orbitals of Os atom and *p* orbitals of carbon atoms. What is more, the other occupied molecular orbitals (HOMO-1 and HOMO-2, shown in Supplementary Fig. 14) also reflect the *d–p* π-electron delocalization. The C=C double bond that is formed combines the $p_\pi$–$p_\pi$ conjugated benzene or pyrene with the $d_\pi$–$p_\pi$ conjugated aromatic metallacycles to form an extended π system containing $d_\pi$–$p_\pi$ conjugation. We also carried out the TD-DFT calculation on the cation of **30**, and found its $\lambda_{max}$ is also 98.5% ascribed to HOMO → LUMO transitions. Both the theoretical spectra of complexes **3** and **30** are consistent with the data from the experimental spectra. The essential reason of the redshifted characteristic energy absorption band is due to the decrease of HOMO–LUMO energy gaps ($E_g$). From DFT calculations, the $E_g$ of compounds **1a**, **3**, and **30** are 2.93, 2.44 and 2.14 eV, respectively (Supplementary Table 8). Furthermore, the decrease

of $E_g$ with further extended *d–p* π-conjugation is mainly due to the introduction of substituents with doner properties that increase the HOMO energy level of the system.

**Applications in organic solar cells (OSCs).** The π-conjugated molecules have been widely utilized for interface engineering of OSCs[46–50]. With such unique $d_\pi$–$p_\pi$ conjugated systems in hand, we explored the performance of these systems as the ETL in OSCs. Devices were fabricated with a conventional structure of ITO/PEDOT:PSS/PM6:Y6/ETLs/Ag (Fig. 6). The photovoltaic performance of devices was studied by employing common interfacial material (PDINO)[51], and the newly synthesized complexes **3** and **30** as ETLs.

As demonstrated in Table 1, the performance of devices based on **3** and **30** were both better than that of the common used ETL (PDINO). A control device with PDINO as ETL without thermal treatment exhibited a PCE of 15.06% which is comparable to the reported value[52,53]. With the extended $d_\pi$–$p_\pi$ π-conjugation, the fill factor (FF) assumed a subsequently increasing trend, and the optimal PCE of OSCs based on **3** and **30** as ETLs were dramatically improved to 15.76% and 16.28% respectively, with enhanced FF and short-circuit current density ($J_{sc}$). To further identify the interface function of these complexes and exclude the effect from the alcohol solvent[54], the performance of devices without ETL or using neat ethanol as the ETL was also investigated (Supplementary Table 3). Both *J–V* curves (Supplementary Fig. 8) displayed lower performance with PCEs of only 10.47% and 11.02%, respectively. At the same time, compared to the performance without ETL, the champion efficiency based on **30** as ETL had an increase of 55%. In addition, to verify the generality of the application of complex **30** in organic solar cells we also tested the results based on the system of PTB7-Th:PC$_{71}$BM (Supplementary Table 5) and PTB7-Th:IEICO-4F (Supplementary Table 6). Both PTB7-Th:PC$_{71}$BM-based device and PTB7-Th:IEICO-4F based device still showed an enhancement of about 10% PCE compared to the PCE of controlled devices. All the *J–V* curves and external quantum efficiency (EQE) spectra were consistent (Figs. 7a, b, Supplementary

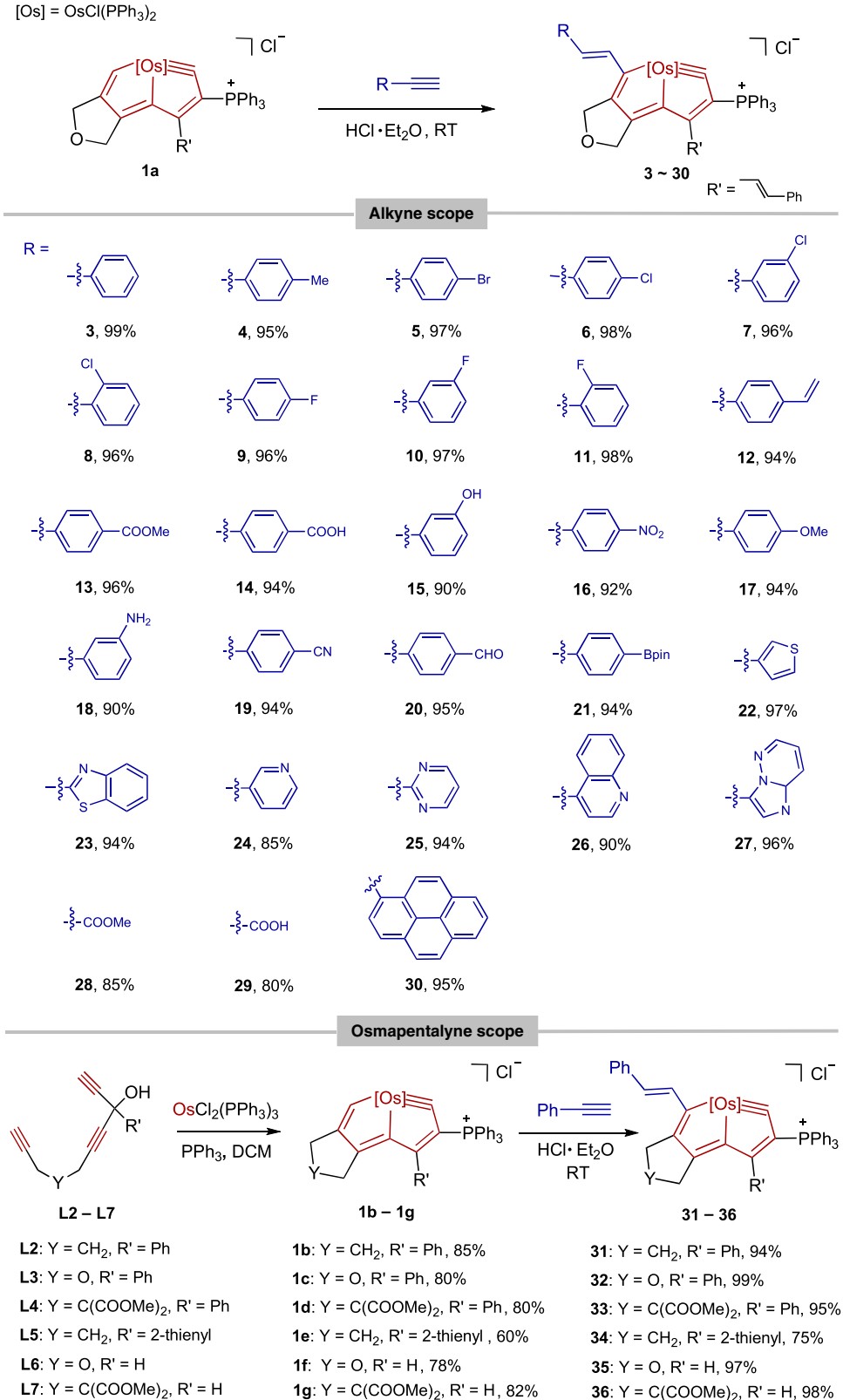

**Fig. 4 Substrate scope.** The substrate scope of the alkynes and osmapentalynes.

Figs. 8–10), and the light response of the device using **30** as ETL was strengthened, thus the $J_{sc}$ was enhanced correspondingly. Therefore, complexes **3** and **30** were demonstrated as effective ETLs with orthogonal alcohol solvent in both fullerene and non-fullerene solar cells to further pump the solar conversions.

To fully understand the effect of different ETLs on carrier transportation, devices using **30** and PDINO as ETLs were studied in parallel. According to the mechanism of OSCs, the normalized absorption spectra of active layer PM6:Y6 spin-coated on **30** and PDINO were tested, respectively (Fig. 7c), revealing

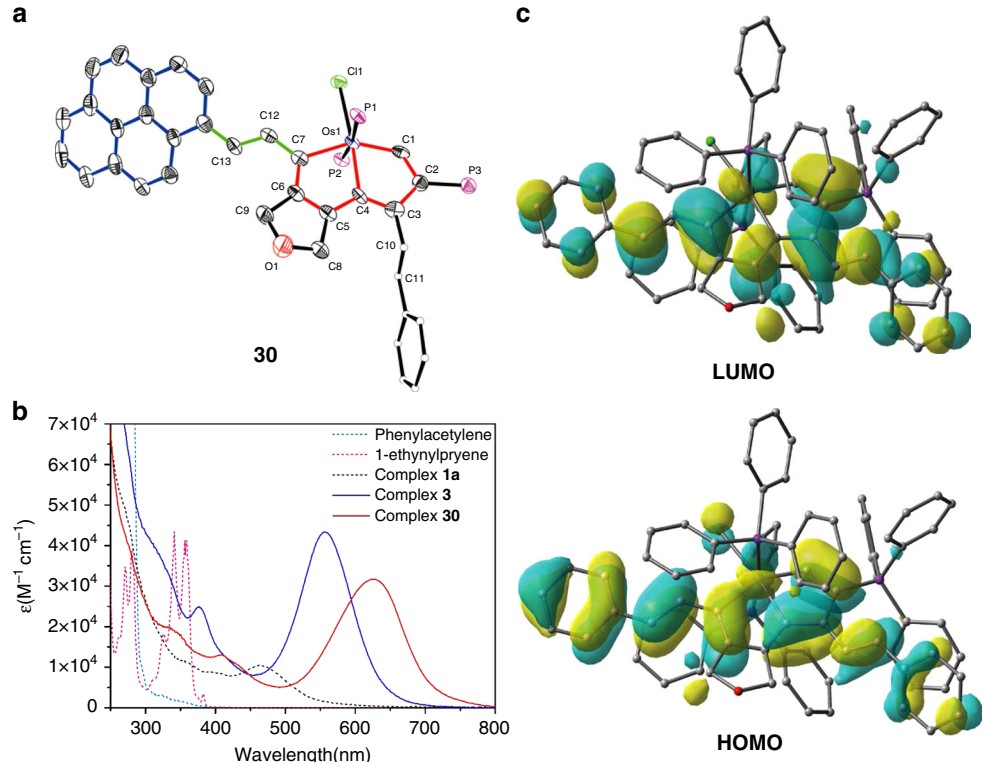

**Fig. 5 Construction of large $d_\pi$–$p_\pi$ conjugated systems. a** X-ray molecular structure for the cation of complex **30**′. Ellipsoids are at the 50% probability level, phenyl groups in PPh₃ are omitted for clarity. **b** UV–Vis absorption spectra of phenylacetylene, 1-ethynylpyrene, complex **1a**, complex **3** and complex **30** measured in DCM at RT (1.0 × 10⁻⁵ M). **c** Selected orbitals related to the excitation of complex **3** (isovalue = 0.02).

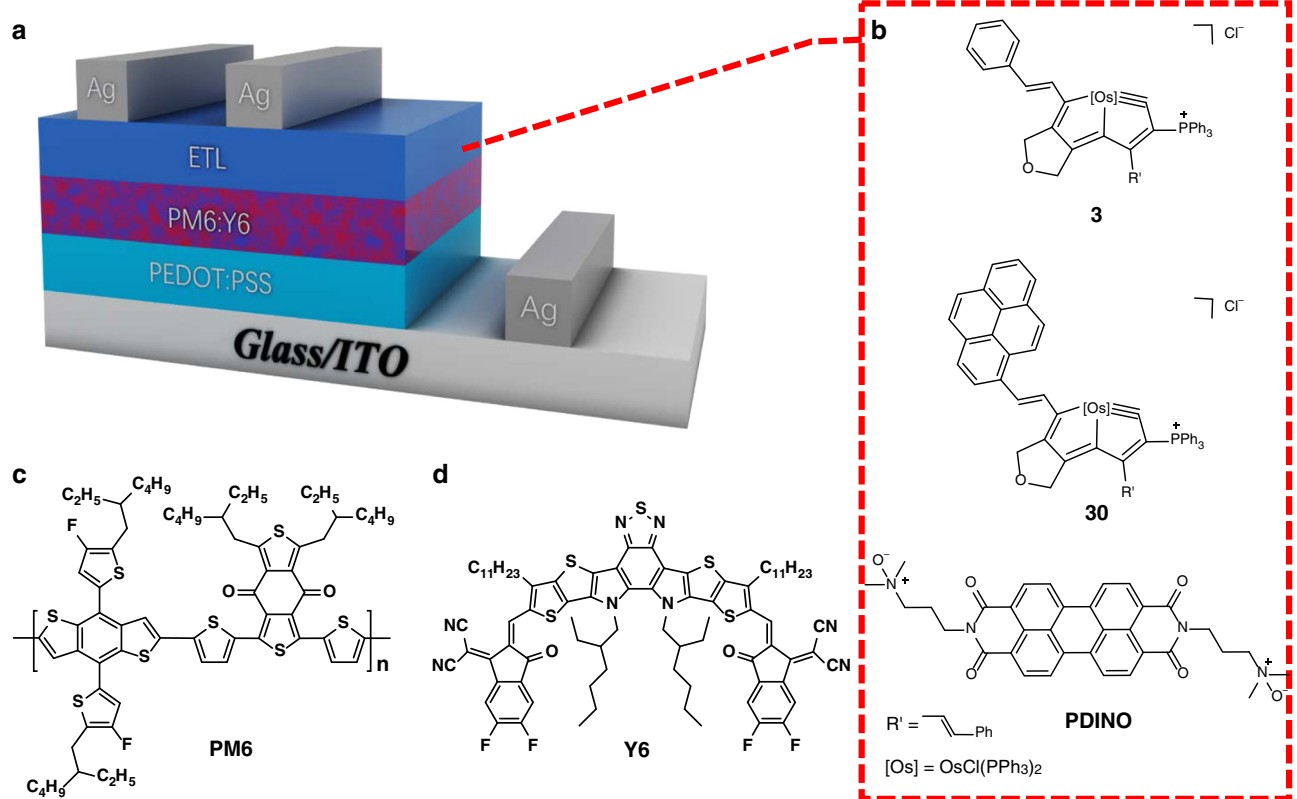

**Fig. 6 Applications in organic solar cells. a** Conventional structure of photovoltaic devices and chemical structures of (**b**) various molecules used as electron-transport layers (ETLs). **c** Polymer donor PM6. **d** Non-fullerene acceptor Y6.

**Table 1 Photovoltaic parameters of PM6:Y6-based OSCs with different ETLs.**

| ETL | $V_{oc}$ (V) | $J_{sc}$ (mA cm$^{-2}$) | FF (%) | PCE (%) | $J_{cal}$[a] |
|-----|------|------|------|------|------|
| PDINO | 0.86 | 24.82 | 70.77 | 15.06 (14.78 ± 0.28)[b] | 24.31 |
| **3** | 0.88 | 24.91 | 72.28 | 15.76 (15.45 ± 0.31) | 24.27 |
| **30** | 0.87 | 25.21 | 74.19 | 16.28 (15.91 ± 0.35) | 24.93 |

[a]The calculated $J_{sc}$ values from EQE curves
[b]Average value ± standard deviation was calculated from the statistics of 20 different devices without thermal treatment

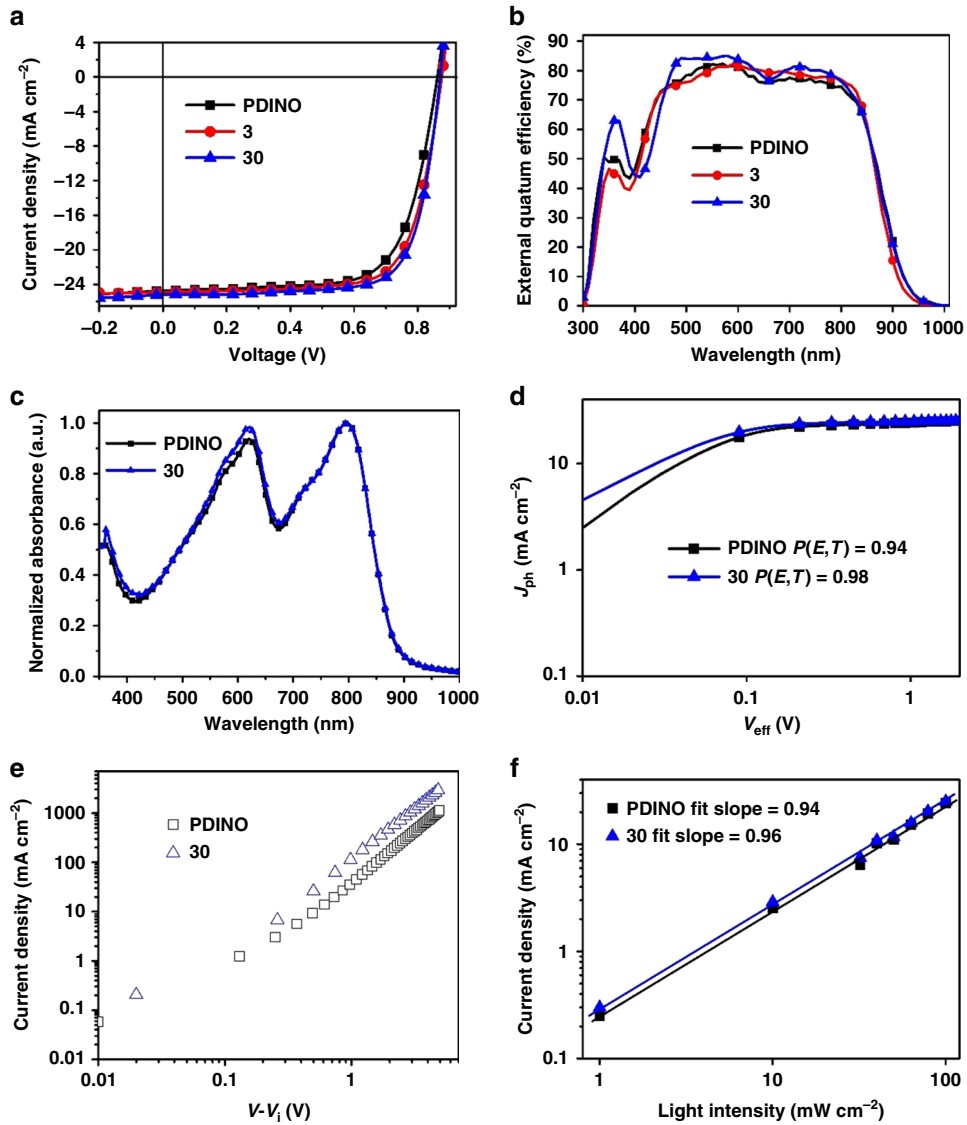

**Fig. 7 The characteristics of organic solar cells devices.** The (**a**) $J$–$V$ and (**b**) EQE curves of OSCs based on PM6:Y6 with different ETLs respectively under 100 mW cm$^{-2}$ AM 1.5 G irradiation. **c** The UV–Vis absorption spectra of blend film PM6:Y6 (**d**) $J_{ph}$–$V_{eff}$ curves. **e** $J$–$V$ curves of electron-only devices. **f** $J_{sc}$ vs Light intensity curves based on PDINO and complex **30**, respectively.

negligible change in the photon absorbance. For the $J_{ph}$–$V_{eff}$ curves plotted in Fig. 7d, the $P(E, T)$ defined as $J_{ph}/J_{sat}$ of a device based on complex **30** increased to 0.98 while the increase with the PDINO-based device was only 0.94, indicating a higher probability of exciton dissociation in the presence of complex **30**. Subsequently, the mobilities of electron-only devices were also calculated in terms of the space-charge-limited current SCLC (Fig. 7e, Supplementary Table 4). Based on complex **30**, the electron mobility ($\mu_e$) was ~5.7×10$^{-4}$ cm$^2$ V$^{-1}$ S$^{-1}$, which is more than three times the PDINO ($\mu_e = 1.7×10^{-4}$ cm$^2$ V$^{-1}$ S$^{-1}$). Meanwhile, dependent on a function of $J_{sc} \propto P$[S55], the fitting slope in Fig. 7f of devices treated by complex **30** was 0.96, which is also higher than that of a PDINO-based device (0.94). These results indicate that device based on complex **30** will inhibit the bimolecular recombination, and eventually will demonstrate a superior device performance than a device based on PDINO.

Therefore, the $d_\pi$–$p_\pi$ π-conjugation systems formed by this newly discovered reaction, as an ETL material, could effectively improve the quality of OSCs, since more efficient carrier transport could be achieved with decreasing carrier recombination and improving carrier mobility.

In summary, we have demonstrated a reaction of M≡C and C≡C bonds that yields acyclic addition products. Based on our experimental observations, together with theoretical calculations, we propose the mechanism of this kind of reaction is the electrophilic addition of carbyne by the alkyne under the synergistic effect of H$^+$. These reactions are highly efficient, regio- and stereospecific, with good functional group tolerance, and are robust under air at RT. Taking advantage of this reaction, two different π-conjugated systems, for example, organic and organometallic systems, can be easily connected through this reaction. The resulting extended π-conjugation systems demonstrate unusually broad and strong absorption in the ultraviolet/visible region. Moreover, they supported impressive applications as ETL materials in OSCs. Specifically, the champion PCE based on complex **30** was enhanced to 16.28%, benefiting from improved carrier transport and restrained carrier recombination as a result of the extended π conjugation. Other properties and applications of such unique $d_\pi$–$p_\pi$ conjugated systems are under further study. In view of the high efficiency under mild reaction conditions and facile experimental operation, this efficient methodology opens significant opportunities for preparation of $d_\pi$–$p_\pi$ conjugated systems for use in functional materials.

## Methods

**General information**. All syntheses of were performed without using standard Schlenk techniques unless otherwise stated. Other reagents were used as received from Aldrich Chemical Co. Column chromatography was performed on alumina gel (200–300 mesh), silica gel (200–300 mesh) or polystyrene gel (Bio-Beads $^{TM}$S-X3 Support, 200-400 mesh) in air. NMR spectra were recorded on a Bruker AVIII-500 ($^1$H, 500.2 MHz; $^{13}$C, 125.8 MHz; $^{31}$P, 202.5 MHz) spectrometer, a Bruker Ascend III 600 ($^1$H, 600.1 MHz; $^{13}$C, 150.9; $^{31}$P, 242.9 MHz) spectrometer at RT, or a Bruker AVIII-400 ($^1$H, 400.1 MHz; $^{13}$C, 100.6 MHz; $^{31}$P, 162.0 MHz) spectrometer at RT. $^1$H and $^{13}$C NMR chemical shifts (δ) are relative to tetramethylsilane, and $^{31}$P NMR chemical shifts are relative to 85% H$_3$PO$_4$. The absolute values of the coupling constants are given in Hertz (Hz). Multiplicities are abbreviated as singlet (s), doublet (d), triplet (t), multiplet (m), and broad (br). High-resolution mass spectra (HRMS) experiments were recorded on a Bruker En Apex Ultra 7.0 T FT-MS. The theoretical molecular ion peak was calculated by Compass Isotope Pattern software supplied by Bruker Co. Elemental analyses were performed on a Vario EL III elemental analyzer. Absorption spectra were recorded on a SHIMADZU UV2550 ultraviolet–visible spectrophotometer. Details for the synthesis and characterization of all the above complexes are given in the Supplementary Information.

**X-ray crystallographic analysis**. All single-crystal X-ray diffraction data were collected on a XtaLAB Synergy, Dualflex, HyPix diffractometer with Cu Kα radiation (λ = 1.54184 Å). With Olex2[56], all the structures were solved using the ShelXT[57] structure solution program using the intrinsic phasing method, and all the structures were refined with the ShelXL[58] refinement package using least-squares minimization. Non-H atoms were refined anisotropically unless otherwise stated. Hydrogen atoms were introduced at their geometric positions and refined as riding atoms unless otherwise stated. All single crystals suitable for X-ray diffraction were grown from a solution of CH$_2$Cl$_2$ layered with hexane. Further details on the crystal data, data collection, and refinements are provided in Supplementary Table 1, Supplementary Table 2, and Supplementary Table 3.

**Computational methods**. All structures were optimized at the B3LYP[59–61] level of functional theory. Frequency calculations were performed to confirm the characteristics of all the calculated structures as minima. All these structures evaluated were optimized at the B3LYP/6–31G*/ED = GD3BJ level of DFT with an SDD basis[62] set to describe Os atoms; single-point energy calculations were then performed on the mechanism using the B3LYP/Def2-TZVP method with the SMD solvation method[63] in DCM. Whereas the UV-Vis-NIR spectrum was used TD-DFT calculations[64] at the B3LYP/6–31G* level of DFT with an SMD solvation model in DCM; an SDD basis set to describe Os atoms. In all calculations, the effective core potentials (ECPs) reported by Hay and Wadt with polarization

functions were added for Os (ζ(f) = 0.886)[65]. All the optimizations were performed with the Gaussian 09 software package[66].

**Experimental methods and characterization of OSCs**. Y6, PTB7-Th was synthesized in our laboratory. PM6 was purchased from the solar material company while PC$_{71}$BM was purchased from the 1-Material Company. First, the ITO substrates were cleaned by ultrasonication. Second, after drying in an oven, these substrates were placed in UV-Ozone for 15 min. Third, a layer of PEDOT:PSS was spin-coated on the surface of ITO at 3000 rpm for 20 s. After annealing the substrates at 150 °C for 15 min and then cooling down, the active layer was prepared by spin-coating PM6:Y6 with D/A = 1:1.1 ratio in chloroform (CF) at 6 mg ml$^{-1}$ with additives 0.5% (v/v) chloronaphthalene (CN). Then the PDINO was spin-coated, complexes **3** and **30** were dissolved in ethanol at 1 mg ml$^{-1}$, at 3000 rpm for 20 s on the active layer as ETLs and subsequently by evaporating the Ag (100 nm) electrode. Through a 0.045 cm$^2$ mask, thermal evaporation in a vacuum chamber requires a pressure of ~3×10$^{-6}$ Pa. All the operations were performed in a nitrogen glove box.

## Data availability

The authors declare that the main data supporting the findings of this study are available within the article and its Supplementary Information file and Supplementary Data 1. The X-ray crystallographic coordinates for structures reported in this study have been deposited at the Cambridge Crystallographic Data Centre (CCDC), under deposition numbers CCDC-1985879 (**1a**), CCDC-1985889 (**2a**), CCDC-1985890 (**3**), CCDC-1985891 (**30**), CCDC-1985896 (**15**), CCDC-1985899 (**29**), CCDC-1985900 (**37**). These data can be obtained free of charge from The Cambridge Crystallographic Data Centre via www.ccdc.cam.ac.uk/data_request/cif. Extra data are available from the corresponding author upon request.

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

## Acknowledgements

This research was supported by the National Natural Science Foundation of China (Nos. U1705254, 21931002, and 21975115), Guangdong Provincial Key Laboratory of Catalysis (No. 2020B121201002), Shenzhen Nobel Prize Scientists Laboratory Project (no. C17783101), and the National Key R&D Program of China (2017YFA0204902). We thank the SUSTech Core Research Facilities for the Holiba-UVISEL measurements.

## Author Contributions

H.X. conceived this project; F.H. conceived the research of organic solar cells; S.C., X.G., L.P., and Y.Z. performed experimental research; S.C. and Y.H. carried out computational studies; L.L. carried out the test experiments of organic solar cells; L.Y. and Y.T. contributed new analytic tools; and S.C., L.L., F.H and H.X. wrote the paper.

## Competing interests

The authors declare no competing interests.
