## [Peer Review File · Nature Communications]

REVIEWER COMMENTS

Reviewer #1 (Remarks to the Author):

Osmapentalynes are a special type of Fischer-carbyne complexes which are surprisingly simple to make. In this report, He, Xia and coworkers continue their studies into structure, reactivity and use of these unusual metalla-aromatic compounds. Specifically, a quite remarkable tautomerization takes place upon protonation of complexes of type 1, which formally shifts the osmium-carbon triple bond from one ring to another. Crystallographic and spectroscopic data prove this reorganization of the pi-system. The newly formed osmapentalyne reacts further with a terminal alkyne to give anti-Markovnikov addition products in excellent yields and selectivity. This step is compatible with many common functional groups. Control experiments show that this reaction does neither proceed via initial [2+2] cycloaddition nor formation of a metal carbene intermediate; rather, it seems that the metal carbyne, assisted by the proton, is able to undergo a direct electrophilic addition to the triple bond, with formation of transient η^2 -complexes which then evolve to the product by restoring the 18-electron count.

For their excellent light absorption properties in the visible range, the resulting products were used for applications in two different types of organic solar cells.

Overall, I think that this paper reports quite unusual new reactivity and leads to products with interesting photophysical properties. Therefore publication is recommended after minor revision:

p. 3: orbitals (not orbits, 2x)

I recommend that the authors comment on what the driving force for the tautomerization of the pi-system from 1 to 2 might be; to me, it is not intuitive and I suppose they may have carried out computational studies

control experiments: when probing the conceivable [2+2] cycloaddition mechanism, it is worth mentioning that terminal alkynes were only recently shown to participate in catalytic alkyne metathesis, see: *Angew. Chem. Int. Ed.* 2012, 51, 13019; *Chem. Eur. J.* 2014, 20, 13188

Computations: it is meaningless to report the computed energies with two digits after the comma (Fig. 3 and main text)

The computed η^2 -complexes (Int2, Int3) bear resemblance to η^2 -complexes of ruthenium that were recently shown to be key intermediates of various unconventional addition reactions to alkynes, see: *JACS* 2018, 140, 3156

substrate scope: while many aryl-alkynes were tested, the only aliphatic alkyne that is reported to give the desired product is propiolic acid (and the derived ester). Which other aliphatic alkynes were tried (but failed to react)? – this info should at least be contained in the SI. Does TMS-acetylene react? Has any internal alkyne been tried: a priori, the proposed mechanism does not exclude internal alkynes

I am not convinced that it is necessary or particularly meaningful to call the new addition process a “click reaction”

Some gentle language polishing is recommended

Reviewer #2 (Remarks to the Author):

This manuscript reports a new reaction pattern between the $C\equiv C$ bond and $M\equiv C$ bond, and their

useful applications in organic solar cells (OSCs). The reaction is highly efficient, regio- and stereospecific, with excellent functional group tolerance, which can also be performed under ambient conditions. The synthesized compounds were adequately characterized by X-ray crystal structure analysis and NMR spectroscopy. Theoretical calculations and the isotope labeling NMR experiments also revealed the mechanism of this reaction. Theoretical aspects of these resulting $d\pi$ - $p\pi$ conjugated systems compounds as well as a range of physical characteristics are explored and discussed. Moreover, the resulting products were found to be good electron transport layer materials in organic solar cells, and the presented device performance was also very impressive, even much higher than that of commonly used electron transport layer, PDINO. This work indicated that this kind of new $d\pi$ - $p\pi$ conjugated complexes had great potential as interfacial layer materials to pump the efficiency of solar devices, especially with their orthogonal solvent processing capability. Therefore, the work presented here is novel and will be of interest to others in the field and the wider community. I recommend it to be published in Nature Communications but only after the following points are addressed.

1. General organometallic reactions require oxygen-free and anhydrous conditions. Why these organometallic reactions can be performed under ambient conditions open to the air?
2. Why the reactivity of non-aromatic substituted alkynes is low? The authors need to explain the details in the main article.
3. The TD-DFT calculations results only showed the main absorption peaks of different compounds (Supplementary Table 6) but didn't show the computational absorption spectra. It would be better to put the experimental and computational absorption spectra of different compounds in figures and should be shown in Supplementary Information.
4. The HOMO and LUMO orbitals of the reactant compound 1a should be shown in the Supplementary Information.
5. Concerned about the application of complex 30 in organic solar cells, besides the system of PM6:Y6 and PTB7-Th:PCBM mentioned in the manuscript, does it work in other systems?

Reviewer #3 (Remarks to the Author):

This paper reports the synthetic methodology to new π -conjugated (ionic) small molecules acting as electron transporting layers in OPVs. There are two aspects of this work equally interesting, the methodology and the applications. At first read, it may appear that the work is disconnected in scope and target, however, after careful evaluation I believe that the two parts work together and the novelty and applicability worth Nature communications upon addressing the following points.

1. The authors should be very clear if acyclic products from this chemistry have never been reported, as stated, or that the pathway to the acyclic products via the cycloaddition intermediate has never been reported. This is a crucial difference.
2. This sentence is unclear "Metal carbynes possess good functional group tolerance for alkynes and are robust in air and under ambient conditions, although these are typical organometallic reactions. Do the authors mean organometallic reactants? If so, why using "typical"? Anyhow, this sentence is confusing.
3. Line 52, page 3. Orbits? What orbits? Orbitals?

4. Same page, replace boost with enhance.
5. page 4, line 68. 1a is one compound with a well-defined structure since in Fig 1a R is well defined. If they want to include generality, they should rewrite the sentence such as "The osmapentalyne reactants such as 1a (Fig 1a) were synthesized...."
6. page 6, lines 98-99. When mentioning click reactions a ref should be included. maybe, the authors could also go further and state that it si also a new click reaction to produce materials useful for OPV and cite the review of Marrocchi and Vaccaro on this front (Chem. Sci. 2016, 7(10), 6298-6308)
7. Page 8 lines 135-137. reaction of 37 does not produce 31'. The authors should provide some details. Do they recover the starting material or what happens?
8. Figure 3. The energy axes should be included in the figure.
9. In the OSC section the film thicknesses should be included, particularly for the ETLs.

Re: Decision on manuscript NCOMMS-20-18495

We express our sincere thanks to the reviewers for their positive and constructive comments on our work. We have taken into account all of the points raised by the reviewers and have highlighted the changes in the revised manuscript. Point-by-point responses to the issues are provided below.

Reviewers' comments:

Reviewer #1 (Remarks to the Author):

Osmapentalynes are a special type of Fischer-carbyne complexes which are surprisingly simple to make. In this report, He, Xia and coworkers continue their studies into structure, reactivity and use of these unusual metalla-aromatic compounds. Specifically, a quite remarkable tautomerization takes place upon protonation of complexes of type 1, which formally shifts the osmium-carbon triple bond from one ring to another. Crystallographic and spectroscopic data prove this reorganization of the pi-system. The newly formed osmapentalyne reacts further with a terminal alkyne to give anti-Markovnikov addition products in excellent yields and selectivity. This step is compatible with many common functional groups. Control experiments show that this reaction does neither proceed via initial [2+2] cycloaddition nor formation of a metal carbene intermediate; rather, it seems that the metal carbyne, assisted by the proton, is able to undergo a direct electrophilic addition to the triple bond, with formation of transient eta²-complexes which then evolve to the product by restoring the 18-electron count.

For their excellent light absorption properties in the visible range, the resulting products were used for applications in two different types of organic solar cells.

Overall, I think that this paper reports quite unusual new reactivity and leads to products with interesting photophysical properties. Therefore publication is recommended after minor revision:

Response: Thank you for the positive comments.

1. p. 3: orbitals (not orbits, 2x)

Response: This is a clerical error. We have corrected it in the revised manuscript.

2. I recommend that the authors comment on what the driving force for the tautomerization of the pi-system from 1 to 2 might be; to me, it is not intuitive and I suppose they may have carried out computational studies

Response: Thank you for your professional suggestions. Actually, the chemistry of tautomerization of the pi-system from **1** to **2** has been discussed in detail in our

previous work (Nat. Commun. **2014**, 5, 3265). The DFT computations have been employed to investigate the mechanism for this conversion. As shown in Figure R1, the tautomerization of the pi-system from **1'** to **2'** undergo the osmapentalene intermediate **3'**. Deprotonation of osmapentalene **3'** at the C1 and C7 carbon atoms only have 12.1 and 7.0 kcal mol⁻¹ free energy reaction barriers, respectively, leading to the formation of osmapentalynes **1'** and **2'**. What's more, the osmapentalyne **2'** is thermodynamically more stable than osmapentalyne **1'**. Following the suggestions, we have added some comments and discussions on this tautomerization to the revised manuscript.

Figure R1. Reaction mechanism for the formation of **1'** and **2'** from **3'**. The computed free energies are in kcal mol⁻¹. (from Nat. Commun. **2014**, 5, 3265)

3. *control experiments: when probing the conceivable [2+2] cycloaddition mechanism, it is worth mentioning that terminal alkynes were only recently shown to participate in catalytic alkyne metathesis, see: Angew. Chem. Int. Ed. 2012, 51, 13019; Chem. Eur. J. 2014, 20, 13188*

Response: Thanks for your comments and constructive suggestions. We have cited the corresponding work in the revised manuscript, please see ref 43 and 44.

4. *Computations: it is meaningless to report the computed energies with two digits after the comma (Fig. 3 and main text)*

Response: Thank you for your suggestions. We have corrected it in the revised manuscript.

5. *The computed eta2-complexes (Int2, Int3) bear resemblance to eta2-complexes of ruthenium that were recently shown to be key intermediates of various unconventional addition reactions to alkynes, see: JACS 2018, 140, 3156*

Response: Thank you for pointing out the resemble conversion chemistry of

eta²-complexes (**Int2**, **Int3**). We have mentioned this work in the revised manuscript to support our mechanism.

6. *substrate scope: while many aryl-alkynes were tested, the only aliphatic alkyne that is reported to give the desired product is propiolic acid (and the derived ester). Which other aliphatic alkynes were tried (but failed to react)? – this info should at least be contained in the SI. Does TMS-acetylene react? Has any internal alkyne been tried: a priori, the proposed mechanism does not exclude internal alkynes*

Response: Thanks for the suggestions. We have tried other aliphatic alkynes, such as, 1-hexyne, cyclopentylacetylene, 6-chloro-1-hexyne, 3-phenyl-1-propyne, propargyl alcohol, propargylamine, 4-pentynenitrile, etc., but none of them could participate in the reaction. The TMS-acetylene can't be reacted either. We have also tried some internal alkynes, such as, diphenylacetylene, 1-phenyl-1-propyne, dimethyl acetylenedicarboxylate, 3-hexyne, etc., but all of them failed to react. Indeed, the low reactivity of non-aromatic substituted alkynes in these reactions mainly due to the following two reasons. Firstly, according to the DFT computations, the reaction mechanism undergoes carbenium ion intermediates. So, the carbenium ion intermediates (**Int2** and **Int3**) can be stabilized by the *sp*² hybrid aromatic groups. Secondly, compared with non-aromatic substituents, the distribution of electrons in π bonds can be activated by the conjugation of aromatic substituted groups, which make them more reactive. As per the suggestions, we have added these information and discussions to the Supplementary Information (page 169).

7. *I am not convinced that it is necessary or particularly meaningful to call the new addition process a “click reaction”*

Response: Thanks for your suggestions. As per the suggestions, we delete the discussion parts of click reactions in the revised manuscript.

8. *Some gentle language polishing is recommended*

Response: As per the suggestions, we have tried our best to improve and modify English. Some gentle language polishing has been done in the revised manuscript.

Reviewer #2 (Remarks to the Author):

This manuscript reports a new reaction pattern between the C≡C bond and M≡C bond, and their useful applications in organic solar cells (OSCs). The reaction is highly efficient, regio- and stereospecific, with excellent functional group tolerance, which can also be performed under ambient conditions. The synthesized compounds were adequately characterized by X-ray crystal structure analysis and NMR spectroscopy. Theoretical calculations and the isotope labeling NMR experiments also revealed the mechanism of this reaction. Theoretical aspects of these resulting $d\pi-p\pi$

conjugated systems compounds as well as a range of physical characteristics are explored and discussed. Moreover, the resulting products were found to be good electron transport layer materials in organic solar cells, and the presented device performance was also very impressive, even much higher than that of commonly used electron transport layer, PDINO. This work indicated that this kind of new $d\pi$ - $p\pi$ conjugated complexes had great potential as interfacial layer materials to pump the efficiency of solar devices, especially with their orthogonal solvent processing capability. Therefore, the work presented here is novel and will be of interest to others in the field and the wider community. I recommend it to be published in Nature Communications but only after the following points are addressed.

Response: Thanks for this referee's positive comments.

1. General organometallic reactions require oxygen-free and anhydrous conditions. Why these organometallic reactions can be performed under ambient conditions open to the air?

Response: Firstly, both the reactants and products show good stability in air atmosphere. Secondly, these reactions are so highly efficient that they can be reacted without oxygen-free and anhydrous conditions. Furthermore, these reactions are thermodynamically favorable from the DFT calculations results (-65.9 kcal mol⁻¹, Figure 3). And the key reaction step (TS1) occurs with small barrier of 18.8 kcal mol⁻¹.

2. Why the reactivity of non-aromatic substituted alkynes is low? The authors need to explain the details in the main article.

Response: The low reactivity of non-aromatic substituted alkynes in these reactions mainly due to the following two reasons. Firstly, according to the DFT computations, the reaction mechanism undergoes carbenium ion intermediates. So, the carbenium ion intermediates (Int2 and Int3) can be stabilized by the sp^2 hybrid aromatic groups. Secondly, compared with non-aromatic substituents, the distribution of electrons in π bonds can be activated by the conjugation of aromatic substituted groups, which make them more reactive. As per the suggestions, we have added these information and discussions to the Supplementary Information (page 169).

3. The TD-DFT calculations results only showed the main absorption peaks of different compounds (Supplementary Table 6) but didn't show the computational absorption spectra. It would be better to put the experimental and computational absorption spectra of different compounds in figures and should be shown in Supplementary Information.

Response: Thanks for your constructive suggestions. Following the suggestions, we have added the figures of the experimental and computational absorption spectra of

different compounds to the Supplementary Information, please see Supplementary Figs. 11 and 12 in Supplementary Information page 14.

Supplementary Figure 11. UV-vis–NIR absorption spectra of carbolong complexes **3** and its modeling fitted one.

Supplementary Figure 12. UV-vis–NIR absorption spectra of carbolong complexes **30** and its modeling fitted one.

4. The HOMO and LUMO orbitals of the reactant compound **1a** should be shown in the Supplementary Information.

Response: Thanks for your valuable suggestions. Following the suggestions, we have added the HOMO and LUMO orbitals of the reactant compound **1a** to the revised Supplementary Information, please see Supplementary Fig. 13 in Supplementary Information page 15.

Supplementary Figure 13. Selected orbitals of complex **1a** showed above (isovalue = 0.02).

5. Concerned about the application of complex **30** in organic solar cells, besides the system of PM6:Y6 and PTB7-Th:PCBM mentioned in the manuscript, does it work in other systems?

Response: Thank you for your suggestions. To verify the generality of the application of complex **30** in organic solar cells, we also tested the results based on the system of PTB7-Th:IEICO-4F. And the results as well showed the improved photovoltaic performance based on complex **30** when it compared with PDINO, which further demonstrated the generality of complex **30** as a material of ETL in organic solar cells. Please see Supplementary Table 6 and Supplementary Figs. 10 in Supplementary Information page 13.

Supplementary Table 6. The performance based on PTB7-Th:IEICO-4F with PDINO/complex **30** as ETL respectively under 100 mW cm⁻² AM 1.5 G irradiation.

ETL	Voc (V)	Jsc (mA/cm ²)	FF (%)	PCE (%)	Jcal ^a
PDINO	0.72	20.40	62.49	9.17 (8.82±0.35) ^b	20.18
Complex 30	0.74	21.72	65.22	10.42 (10.09±0.33)	21.35

^aThe calculated *Jsc* values from EQE curves;

^bAverage value ± standard deviation were calculated from the statistics of 20 different devices

Supplementary Figure 10. The (A) J-V and (B) EQE curves of OSCs based on PTB7-Th:IEICO-4F with PDINO/complex 30 as ETL respectively under 100 mW cm⁻² AM 1.5 G irradiation.

Reviewer #3 (Remarks to the Author):

This paper reports the synthetic methodology to new pi-conjugated (ionic) small molecules acting as electron transporting layers in OPVs. There are two aspects of this work equally interesting, the methodology and the applications. At first read, it may appear that the work is disconnected in scope and target, however, after careful evaluation I believe that the two parts work together and the novelty and applicability worth Nature communications upon addressing the following points.

Response: Thank the reviewer for the positive comments.

1. The authors should be very clear if acyclic products from this chemistry have never been reported, as stated, or that the pathway to the acyclic products via the cycloaddition intermediate has never been reported. This is a crucial difference.

Response: Thanks for your comments. We have done extensive and detailed literature research. To the best of our knowledge, the acyclic products from the reaction of M≡C and C≡C bonds have indeed never been reported.

2. This sentence is unclear "Metal carbynes possess good functional group tolerance for alkynes and are robust in air and under ambient conditions, although these are typical organometallic reactions. Do the authors mean organometallic reactants? If so, why using "typical"? Anyhow, this sentence is confusing.

Response: Thank you for your suggestions. This is incorrect usage of "typical". We mean that "In contrast to common organometallic reactions that are sensitive to air and temperature, metal carbynes possess good functional group tolerance for alkynes and are robust in air and under ambient conditions". We have corrected it in the revised manuscript.

3. Line 52, page 3. *Orbits? What orbits? Orbitals?*

Response: This is a clerical error. It means orbitals. We have corrected it in the revised manuscript.

4. *Same page, replace boost with enhance.*

Response: Thank you for your suggestions. We have corrected it in the revised manuscript.

5. *page 4, line 68. 1a is one compound with a well-defined structure since in Fig 1a R is well defined. If they want to include generality, they should rewrite the sentence such as "The osmapentalyne reactants such as 1a (Fig 1a) were synthesized...."*

Response: Thank you for nicely pointing out the typos or incorrect usage of some words. The mentioned errors have been corrected in the revised manuscript.

6. *page 6, lines 98-99. When mentioning click reactions a ref should be included. maybe, the authors could also go further and state that it is also a new click reaction to produce materials useful for OPV and cite the review of Marrocchi and Vaccaro on this front (Chem. Sci. 2016, 7(10), 6298-6308)*

Response: Thank you for your suggestions. These reactions are highly efficient and specific, which satisfy the characteristics of the click reactions. So it is indeed necessary to cite the reference when mentioned click reactions. However, one of the reviewers can't be convinced that it is necessary or particularly meaningful to call the new addition process a "click reaction". Therefore, in order to avoid controversy, we deleted the discussion parts of click reactions in the revised manuscript.

7. *Page 8 lines 135-137. reaction of 37 does not produce 31'. The authors should provide some details. Do they recover the starting material or what happens?*

Response: Thank you for your suggestions. Compound 37 can't be reacted with phenylacetylene, and they recover the starting material. We have corrected it in the revised manuscript.

8. *Figure 3. The energy axes should be included in the figure.*

Response: Following the suggestions, we have added the energy axes in the Figure 3.

9. *In the OSC section the film thicknesses should be included, particularly for the ETLs.*

Response: Thank you for your suggestions. The film thicknesses of active layer have supported in Supplementary Table 4. Particularly, as recommend, we have supplemented the thickness of ETLs measured by the Spectroscopic Ellipsometer in following table and figure, and we have added it in Supplementary Table 4.

Table R1. The thickness of PDINO/complex **30** on silicon.

ETL	Thickness (nm)
PDINO	7.519±0.062
Complex 30	6.972±0.328

Figure R2. The whole optical spectrum fitting curves of silicon basement with PDINO/complex **30**.

REVIEWERS' COMMENTS:

Reviewer #1 (Remarks to the Author):

After carefully studying the revised version of this manuscript, I conclude that the authors have addressed all major issues raised by the reviewers in the first round of refereeing. Appropriate modifications to the manuscript and the SI have been made. I recommend publication as it stands.

Reviewer #2 (Remarks to the Author):

The authors have revised the manuscript by successfully addressing the reviewers' comments. Therefore, I recommend its publication as it is.

Reviewer #3 (Remarks to the Author):

The authors have addressed all my suggestions as well as of the other reviewers. This paper can be published as is, in my opinion.

A point-by-point response to issues raised by the referees is summarized below.

REVIEWERS' COMMENTS:

Reviewer #1 (Remarks to the Author):

After carefully studying the revised version of this manuscript, I conclude that the authors have addressed all major issues raised by the reviewers in the first round of refereeing. Appropriate modifications to the manuscript and the SI have been made. I recommend publication as it stands.

Response: Thank the reviewer for the positive comments.

Reviewer #2 (Remarks to the Author):

The authors have revised the manuscript by successfully addressing the reviewers' comments. Therefore, I recommend its publication as it is.

Response: Thank the reviewer for the positive comments.

Reviewer #3 (Remarks to the Author):

The authors have addressed all my suggestions as well as of the other reviewers. This paper can be published as is, in my opinion.

Response: Thank the reviewer for the positive comments.

We have taken into consideration all of the reviewers comments and criticisms. Finally, we would like to extend our thanks again to all of the reviewers for their helpful suggestions and comments.

With kind regards and best wishes,

Haiping Xia